# A Review of the Tear Film Biomarkers Used to Diagnose Sjogren’s Syndrome

**DOI:** 10.3390/ijms251910380

**Published:** 2024-09-26

**Authors:** Jason Peng, David Feinstein, Salvatore DeSimone, Pietro Gentile

**Affiliations:** 1Cooper Medical School of Rowan University, Camden, NJ 08103, USA; pengja23@rowan.edu; 2Department of Rheumatology, Cooper University Hospital, Camden, NJ 08103, USA; feinstein-david@cooperhealth.edu (D.F.); gentile-pietro@cooperhealth.edu (P.G.); 3Department of Ophthalmology, Cooper University Hospital, Camden, NJ 08103, USA

**Keywords:** Sjogren, dry eye, tear, lacrimal gland, biomarker

## Abstract

This literature review looks at Sjogren’s Syndrome (SS), a chronic autoimmune disorder affecting exocrine glands, particularly the lacrimal and salivary glands. SS manifests as ocular and oral dryness, with severe complications like visual dysfunction and corneal perforation, as well as systemic implications, such as interstitial lung disease and lymphoma. This review explores the use of tear film biomarkers to diagnose SS, emphasizing the significance of their identification in aiding clinical diagnosis and differentiation from other diseases. This study identified and analyzed 15 papers, encompassing 1142 patients and employing various tear sample collection methods. Tear biomarkers were categorized by function and explored in-depth. Categories include (1) antimicrobials, antivirals, and antifungals; (2) components of immune regulation; (3) components that regulate metabolic processes; and (4) inflammatory markers. Noteworthy findings include the potential diagnostic values of tear lysozyme, lactoferrin, dinucleoside polyphosphates, cathepsin, defensin, antibodies, epidermal fatty acid-binding protein, HLA-DR, ADAM10, aquaporin 5, and various miRNAs and mRNAs. Overall, our understanding of SS tear film composition is enhanced, providing valuable insights into the pathogenesis of SS and offering a foundation for future diagnostic and therapeutic advancements in autoimmune conditions affecting the ocular surface.

## 1. Introduction

Sjogren’s syndrome (SS) is a chronic autoimmune disorder that is marked by lymphocytic infiltration into the exocrine glands throughout the body and their consequent inflammation and destruction [1,2]. This can either be caused by another underlying rheumatological disease (secondary; sSS) or not (primary; pSS) [1,2]. The lacrimal and salivary glands are the main ones affected, resulting in the disorder’s hallmark symptoms of ocular and oral dryness [1,2]. This is serious because SS-related dry eye is connected to visual dysfunction and, subsequently, corneal perforation, uveitis, scleritis, retinal vasculitis, and optic neuritis [3]. Furthermore, dry mouth can make tasting, chewing, and swallowing more difficult, as well as increase the risks of contracting cavities, tooth loss, and oral infections [4]. SS also has a research-backed association with interstitial lung disease, non-Hodgkin lymphoma, irritable bowel syndrome, kidney disease (mainly tubulointerstitial nephritis), and vasculitis [5,6,7,8,9]. In the United States alone, SS affects up to 4 million individuals of all races and ethnicities, but predominantly those that are over 50 years old and of the female sex (9:1 female-to-male) [4,10].

Another characteristic of SS is B-cell hyperactivity, which triggers hypergammaglobulinemia and the formation of serum autoantibodies [2]. Antinuclear antibodies and immunoglobulins formed against rheumatoid factor, cryoprecipitate, Ro/SSA, and La/SSB are key biomarkers of the disease [2]. While these antibodies are good diagnostic indicators of SS, due to their high serum availability and their formation by the human body well before symptoms start to appear, collecting serum samples can sometimes be difficult and complex [11,12]. Moreover, due to significant advancements in profiling technologies’ sensitivity over the past decade, even bodily fluids with minute volumes are now viable for biomarker identification [12]. Tear films, in particular, have emerged as a convenient and less invasive source of biomarkers [12]. They are being extensively studied through proteomics, lipidomics, and metabolomics analyses for this purpose [12].

This is especially relevant because dry eye, a prominent manifestation of SS, shares many features with dry eye disease (DED), a condition in which the tear film is insatiable and hyperosmolar, resulting in inflammation and damage to the ocular surface [12,13]. The underlying mechanisms for both SS and DED are rooted in lacrimal gland dysfunction [12,13]. In SS, there is an autoimmune attack on the lacrimal glands, decreasing tear secretion and subsequently compromising tear film stability [12,13]. Emerging research shows complex interactions between the microbiome of the ocular surface and components of tear fluid that play a critical role in the pathogenesis of DED. Specific bacteria, including Corynebacterium tuberculostearicum in conjunctival samples and Propionibaceriaceae in eyelid samples, have also been found more abundantly in DED patients in comparison to healthy individuals. Furthermore, functional differences in this microbiome, namely enriched biosynthesis pathways in conjunctival samples from healthy patients, highlight the potential for microbial and protein markers in understanding DED pathology [12,13]. This link between SS and DED underscores the importance of evaluating tear film biomarkers, as they can provide insights into the pathophysiology of SS-induced dry eye and offer potential diagnostic and therapeutic targets.

One objective of this systematic literature review will be to explore which biomarkers can be found in the tear films of SS patients in order to create a uniform list that clinicians can use to aid in their diagnosis of SS and differentiation from other diseases. While past reviews have been conducted for similar diseases like DED, an in-depth, a comprehensive review of the tear biomarkers for both pSS and sSS, specifically, has not yet been accomplished [12,13].

In general, the tear film is a complex structure that maintains the health and function of the ocular surface [14]. It retains the moisture of the eye’s surface, provides lubrication during blinking, and maintains a clear optical surface for vision [14]. The various proteins, enzymes, and nutrients that it contains protect against microbial infections and help repair ocular tissues [14].

A deeper understanding of tear films is required to understand where these biomarkers come from. Tear film is a ~3 μm thick bi-fluid layer that is responsible for protecting, lubricating, and maintaining a smooth surface for light refraction over the ocular surface, as well as preserving the conjunctiva and avascular cornea [14]. It consists of three layers—the inner mucin, middle aqueous, and outer lipid layers—all of which slightly overlap and mix [14]. Many of the biomarkers we will be looking at are proteins found within the mucin/aqueous layers (often combined because of their ambiguous borders), which contain fluid and soluble factors, made by the lacrimal glands, and mucins, made by the goblet cells [15]. As such, this paper will also discuss the link between lacrimal gland inflammation and insufficiency by proposing the most likely mechanisms of action connecting the two and determining how this causes the tear film’s composition to change. While other papers have proposed ideas regarding this in the past, this will be an up-to-date review of the available research.

## 2. Materials and Methods

A literature search for the identification of relevant studies was conducted on 1 August 2023, using the following electronic databases:

PubMed (https://pubmed.ncbi.nlm.nih.gov/, accessed on 1 August 2023);

Ovid MEDLINE (accessed on 1 August 2023).

The context of this review was to evaluate the current literature in order to identify the biomarkers for SS that are found in tear film and the underlying mechanism of action behind the change in tear composition. The search strategy principle was to divide the search into multiple, grouped terms. PubMed was first searched to determine if the MeSH terms produced articles that accurately fit the topic of this study. This was also carried out to find corresponding keyword equivalences to increase the sensitivity of the literature search. The search terms included the following: “sjogren”, “sjogren’s syndrome”, “autoimmune”, “autoimmune disease”, “tear”, “eye”, “ophthalmic”, “ocular”, “ocular manifestations”, “biomarker” and “pathophysiology”. “Sjogren’s syndrome”, “Autoimmune disease” and “Ocular manifestations” were all removed for being redundant with “Sjogren”, “Autoimmune” and “Ocular”, respectively. These terms were paired as the following: ‘sjogren, tear’, ‘sjogren, eye’, ‘sjogren, ophthalmic’, ‘sjogren, ocular’, ‘sjogren, biomarker’, ‘sjogren, pathophysiology’, ‘autoimmune, tear’, ‘autoimmune, eye’, ‘autoimmune, ophthalmic’, ‘autoimmune, ocular’, ‘autoimmune, biomarker’, and ‘autoimmune, pathophysiology’. The total number of articles accumulated was 24,090, which was then reduced to 8599 when removing duplicates. After looking through roughly 10–20 articles from each paired search term, it was determined whether or not the article topics clearly matched the topic of our paper. ‘Sjogren, pathophysiology’ and ‘autoimmune, pathophysiology’ were removed for being irrelevant to the topic. ‘Autoimmune, tear’, ‘autoimmune, eye’, ‘autoimmune, ophthalmic’, ‘autoimmune, ocular’ and ‘autoimmune, biomarker’ were all removed for being too general, including many papers on diseases that were not SS. This left 1149 papers.

The remaining grouped terms were then combined and coded into PubMed and Ovid MEDLINE as the following: (“Sjogren” [All Fields] AND (“tear” [All Fields] OR “eye” [All Fields] OR “ophthalmic” [All Fields] OR “ocular” [All Fields]) AND “biomarker” [All Fields]). This resulted in 43 papers. None of these papers were eliminated after filters were applied, including (1) Humans only, (2) Publication Dates starting from 1 January 2010, and (3) English only. Then, 13 of these were removed because the biomarkers identified were not isolated from the tear film (collected from saliva, serum, or genetic testing). A further 4 papers were removed, as they were related to diseases other than SS, 2 were removed for only looking at animal models, and 2 were removed for only studying pediatric populations (birth to 18 years old). The remaining 22 papers were screened by examining the reference lists of all the articles they included. Subsequently, 7 papers were removed for being secondary research sources (reviews, systematic reviews, and meta-analyses). Note that these secondary sources are still cited to support points made by original research sources in the discussion.

## 3. Results

The 15 papers selected for the final analysis were reviewed for their methodologies on tear film biomarker identification. Refer to Figure 1 for the screening process via the PRISMA flow diagram. We assessed each study based on the techniques they used to collect and analyze tear samples, as well as the criteria for identifying biomarkers. Prevalence data for each biomarker were extracted by calculating the percentage of patients in the respective studies who demonstrated elevated or detectable quantities of said biomarker. The biomarker data were aggregated across studies by calculating the relative prevalence of each tear film biomarker in relation to the total number of patients reported in each study, dividing the number of patients showing the biomarker in each study by the total number of patients examined for that specific biomarker, and summarizing it across the studies. Additionally, the function and pathophysiological mechanisms of each biomarker were reviewed, focusing on how they contributed to tear film dysfunction in SS. For example, matrix metalloproteinases were analyzed for their roles in inflammation and tissue breakdown, while the discussion on the dysregulation of cytokines like IL-1a and IL-6 focused on their involvement in autoimmune-mediated lacrimal gland dysfunction. 

The unique role that each biomarker plays in SS symptomatology, in comparison to other autoimmune or ocular conditions, was also analyzed where data permitted. Doing so elucidated the specificity of tear film biomarkers for SS. This study was limited by variability in the methodologies used across the included studies, including differences in sample collection techniques and assay sensitivities.

## 4. Discussion

### 4.1. Studies Included

A total of 15 papers regarding the use of tear biomarkers for the diagnosis of SS (either pSS, sSS, or both) and differentiation from other similar diseases, as well as how the underlying mechanism of disease causes changes to the tear composition, were analyzed. These included 6 cross-sectional, 2 case–control, 2 comparative, 1 cross-section case–control (identified as both), 1 retrospective, 1 cohort, 1 experimental, and 1 case series study over 7 countries. In total, 1142 patients were analyzed (434 healthy controls, 162 with DED but without any comorbid condition, 534 with pSS-DED, and 12 with sSS-DED). The tear samples were collected using either Schirmer strips (59.1%), capillary/microcapillary tubes (27.3%), micropipettes/automated pipettes (13.6%), filter paper discs (4.5%), polyester wicks (4.5%), or lateral flow immunoassay test strips (4.5%). The biomarkers were separated into the categories listed below, based on function. 

When available, the following information will be provided for each biomarker type: (a) what the biomarker is and where is it found, (b) what mechanism of action causes its increase or decrease in SS, and (c) extraneous details that are relevant to the topic. The key characteristics of each study can be found in Table 1, below. Extraneous study characteristics and limitations for each study can be found in Appendix A. 

### 4.2. Antimicrobials, Antivirals, and Antifungals

#### 4.2.1. Lysozyme

Lysozyme is a bacteriolytic enzyme that is secreted by the main and accessory lacrimal glands and is found in various body fluids, including tears, where it plays a role in protecting the ocular surface against pathogens [16,17]. It was found to be downregulated in SS patients with DED, potentiating ocular surface infection [16]. In fact, this is likely why patients with SS-associated DED have more ocular surface infections than those with MGD-associated DED [17]. The decrease in lysozyme levels may also be linked to the ongoing inflammatory processes and immune dysregulation in SS. The loss of lysozyme’s protective function on the ocular surface may further exacerbate dry eye symptoms and contribute to the overall instability of the tear film. Overall, a drop in the content of tear lysozyme is an effective diagnostic marker to diagnose SS and differentiate it from MGD [17]. It is important to note that lysozyme C, specifically, is a biomarker that overlaps with SS, DED, and MGD [18].

#### 4.2.2. Lactoferrin

Lactoferrin (Lf) is an iron-binding glycoprotein with antimicrobial properties found in various secretions, including tears [19,20]. One meta-analysis reported a significantly decreased concentration of Lf in the tears of patients with DED, which often occurs concomitantly with SS [19]. Another study found that Lf concentration specifically decreases in the tears of patients with SS [20]. Because of these slightly differing conclusions, Lf should be used as a supportive diagnostic biomarker for DED and SS, but should be used in tandem with other markers.

Like lysozyme, the downregulation of lactoferrin may compromise the ocular surface’s antimicrobial defense mechanisms and contribute to the increased susceptibility to infections in SS. The decrease in lactoferrin levels may be linked to the overall inflammatory milieu on the ocular surface and immune dysregulation in SS. The loss of lactoferrin’s protective function on the ocular surface may further contribute to tear film instability and dry eye symptoms in SS.

#### 4.2.3. Dinucleoside Polyphosphates

Dinucleoside polyphosphates (NpnN) are naturally occurring substances found in tears, aqueous humor, and the retina, crucial for maintaining ocular surface health [21]. They stimulate tear secretion, promote mucin release from goblet cells, accelerate cell migration for wound healing, and enhance ocular defense by stimulating the formation of antimicrobial proteins like lysozyme and lactoferrin [21]. The compound Ap4A, elevated in patients with dry eye, Sjogren syndrome, or congenital aniridia, or after refractive surgery, shows promise as a potential biomarker for dry eye conditions and could aid in diagnosing SS and differentiating it from similar diseases [21]. However, further research is needed to fully understand their diagnostic and therapeutic potentials, especially regarding diadenosine polyphosphates’ elevation in glaucoma patients and their impact on intraocular pathophysiology [21]. Exploring the underlying mechanisms of dinucleoside polyphosphates could lead to exciting advancements in SS diagnosis and treatment, offering valuable insights into ocular surface disorders.

#### 4.2.4. Cathepsin and Cystatin

Cathepsin S (CatS) has emerged as a potential biomarker for the diagnosis of both pSS and sSS. In the present study, tear CatS (CTSS) activity was found to be significantly elevated in the tears of SS patients, compared to healthy controls, and in patients with non-SS related dry eye and other non-SS autoimmune diseases, suggesting its potential as a novel biomarker for SS diagnosis [22]. Notably, the elevated CTSS activity in SS patients was not solely a result of small tear volumes, as it remained higher regardless of the volume of tears produced. Interestingly, tear CTSS activity was found to be equally elevated in both pSS and sSS patients, indicating its potential for use in the diagnosis of either type of SS [22,23]. While CTSS was statistically more robust in distinguishing SS patients from general autoimmune disease patients, rather than sicca or non-specific dry eye patients, further investigation is necessary to fully assess its utility. Moreover, it is noteworthy that additional biomarkers are also under investigation for potential usefulness in SS diagnosis, including tear Cathepsin S [24].

The underlying mechanism of increased tear CTSS activity in SS patients appears to involve alterations in the balance between proteases and their inhibitors. Cystatin C (Cys C), an endogenous inhibitor of CTSS activity, was found to be significantly reduced in the tears of SS patients compared to patients with other autoimmune diseases and non-autoimmune dry eye, which may contribute to the elevated CTSS activity seen in SS tears [23,25]. This reduction in Cys C may lead to the partial degradation of other tear proteins, including Lf and sIgA, which were also found to be decreased in SS tears [25]. It is postulated that a lack of protective factors, potentially including Cys C or other protease inhibitors, might prevent CTSS-mediated proteolytic degradation in SS patient tears [25]. Pro-inflammatory cytokines implicated in SS, such as interleukin-6 and interferon-γ, downregulate Cys C expression and secretion in immune cells, further contributing to the imbalance of proteases and inhibitors in SS tears [25].

The findings from this study shed light on the potential utility of CTSS activity and Cys C levels in tears as biomarkers for SS diagnosis and disease activity assessment [25]. Notably, these tear biomarkers showed equal abilities to identify both primary and secondary SS patients, which is particularly relevant, as most established SS biomarkers focus on primary SS patients [26]. Elevated CTSS activity was observed to a lesser extent in non-autoimmune dry eye patients, suggesting its potential as a marker to differentiate SS from other dry eye conditions [25]. Mechanistically, the study proposes that increased tear CTSS activity in SS may result from decreased Cys C levels and possible deficiencies in additional endogenous protease inhibitors [25]. The imbalance of proteases and inhibitors may lead to the degradation of other tear proteins, contributing to the overall changes in tear composition observed in SS patients [25]. Combining tear CTSS and Lf measurements may further enhance the ability to distinguish SS patients from non-autoimmune dry eye patients [25]. Overall, these findings contribute to the understanding of tear biomarkers and their underlying mechanisms in SS, providing insights into the diagnosis and differentiation of this autoimmune condition [25].

#### 4.2.5. Defensin

Defensins are secreted proteins with broad-spectrum activity against viruses, bacteria, and fungi [16]. Their production was found to be upregulated in patients with SS-associated DED [16]. This dysregulation shows the involvement of the inflammatory response in autoimmune-mediated diseases like SS [16].

### 4.3. Immune Regulation

#### 4.3.1. Antibodies

Antibodies against specific autoantigens have been studied as potential biomarkers for SS diagnosis and disease activity. The presence of anti-Ro/SSA and anti-La/SSB antibodies in the tears of SS patients was investigated and a correlation was found between their presence in serum or tear fluid and the severity of keratoconjunctivitis sicca [27]. Additionally, antibodies against a-fodrin have been detected in tear fluid samples from patients with SS, and their levels were correlated with the severity of eye involvement [27]. The presence of these autoantibodies in tears suggests an immune-mediated attack on the ocular surface, contributing to the pathogenesis of dry eye symptoms in SS [27].

The detection of specific autoantibodies in tears may serve as a diagnostic marker for SS and provide valuable clinical information regarding concurrent diagnosis and severity. More research is required to understand the roles of these antibodies in the immune response and their interactions with ocular surface tissues.

#### 4.3.2. E-FABP

Epidermal fatty acid-binding protein (E-FABP) is a member of the fatty acid-binding protein family, which acts as epithelial barriers for the ocular surface and trans-epithelial water transporter [28]. This one in particular regulates immunity, balancing Th17 and Treg cell populations by acting as peroxisome proliferator-activating receptor γ ligands via the metabolic–inflammatory pathway [28]. It is produced by lacrimal, sebaceous, and meibomian glands, and is found mainly on the ocular surface epithelium [29]. E-FABP is known to bind to free fatty proteins and downregulate the genes of inflammatory markers, specifically those for Nos2, Cxcl10, and IL-6 [28]. This means that, in theory, E-FABP levels should decrease when inflammatory pathways activate [28].

Study findings confirmed this, as, while E-FABP does not have significant differences in saliva or serum, there was a lower concentration of it in SS patients when compared to healthy individuals [28]. This suggests that this biomarker can be used to assess epithelial damage of the ocular surface, as it is correlated with decreased tear stability and quality [28]. The downregulation of E-FABP may imply altered lipid metabolism and transport on the ocular surface in SS.

#### 4.3.3. HLA-DR

Human leukocyte antigen (HLA) molecules play a crucial role in the immune response by presenting antigens to T cells [30]. In SS, the expression of HLA-DR has been extensively studied as both a potential biomarker for early detection and as a marker for monitoring disease activity. An increased HLA-DR expression in the conjunctival cells of SS patients compared to controls was discovered [30]. The upregulation of HLA-DR may indicate the presence of activated antigen-presenting cells and immune activation in the ocular surface of SS patients.

The overexpression of HLA-DR may contribute to the pathogenesis of SS by facilitating the presentation of self-antigens to autoreactive T cells, leading to an autoimmune attack on exocrine glands. Additionally, HLA-DR expression may serve as an indicator of disease severity and activity in SS, providing valuable clinical information for disease management [30].

### 4.4. Metabolic Processes

#### 4.4.1. Proteolytic Cleavage

A disintegrin and metalloproteinase domain-containing protein 10 (ADAM10) is a transmembrane protein known to cleave cell surface proteins, including cytokines, receptors, and adhesion molecules, thus modulating various cellular processes [31]. In the context of SS, ADAM10 has been identified as a potential biomarker in tears. Increased levels of ADAM10 were observed in the tears of both pSS and sSS patients, compared to healthy controls [31]. The upregulation of ADAM10 suggests its involvement in tear film instability and inflammatory processes in SS.

ADAM10’s role in proteolytic cleavage of cell surface proteins implies its potential influence on cytokine and receptor signaling, which may contribute to the immune dysregulation observed in SS. Specifically, this protease may regulate the release of TNF-α and IL-6R inflammatory cytokines [31]. The identification of ADAM10 as a dysregulated tear protein may offer new insights into the pathogenesis of SS and provide a novel diagnostic marker for distinguishing SS from other similar diseases.

#### 4.4.2. Secretion

Aquaporin 5 (AQP5) is a water channel protein expressed in the lacrimal and salivary glands, playing a critical role in tear and saliva secretion [27]. Increased AQP5 levels in the tears of SS patients, in comparison to healthy controls, was reported [27]. The increased levels of AQP5 may directly impact tear secretion and lead to the characteristic dry eye symptoms observed in SS.

The increase in AQP5 levels in tears may be attributed to the autoimmune attack on the lacrimal glands, leading to a potential compensatory increase in production as tear production is decreased. Additionally, the altered expression of AQP5 may reflect the overall inflammatory milieu in the ocular surface of SS patients, further affecting the homeostasis of the tear film.

#### 4.4.3. Ubiquitination

The literature review revealed that tear biomarkers associated with ubiquitination are altered in patients with primary pSS. Through proteomic analysis using Scaffold, the five most upregulated proteins in tear fluid from pSS patients were identified, and among them were proteins involved in ubiquitination, including LIM domain only protein 7 (LMO7). Further quantitative analysis demonstrated significantly higher levels of LMO7 in tear fluid from pSS patients expressing pathological dry eye disease (DED) manifestations, compared to healthy controls. This suggests that ubiquitination-related pathways may play roles in the underlying mechanisms of disease, possibly affecting cell signaling, cell adhesion, and immune responses [32,33,34].

#### 4.4.4. Neutrophil Degranulation

The literature review identified proteins linked to neutrophil degranulation as potential biomarkers for differentiating pSS patients from healthy controls. Erythrocyte band 7 integral membrane protein (STOM) was significantly unregulated in pSS patients, indicating its role in the activation of neutrophil. The elevated levels of Annexing A4 (ANXA4) and Annexing A11 (ANXA11) in pSS patients suggest they may play roles in regulating neutrophil activation and degranulation. These findings indicate that neutrophil-related pathways may be altered in pSS, providing insights into diagnosis and differentiation from similar diseases [33].

#### 4.4.5. Cell Differentiation

The literature review identified tear protein D52 (TPD52) as an upregulated biomarker involved in B cell differentiation in pSS patients. This protein may play a central role in regulating B cell differentiation processes, which are crucial in the pathogenesis of autoimmune diseases like pSS. Additionally, tear protein D52 (TPD52) and E3 ubiquitin-protein ligase HUWE1 (HUWE1) were found to be upregulated in pSS patients, potentially influencing cell differentiation processes. Understanding these pathways may provide valuable insights into the molecular mechanisms underlying pSS and its differentiation from other similar diseases [33].

#### 4.4.6. Calcium Signaling (Ca+2 Signaling)

Tear biomarkers related to calcium signaling pathways were also identified in pSS patients. Copine (CPNE1) was significantly upregulated in tear fluid from pSS patients and is involved in TNF-α receptor signaling, inflammation, and apoptosis. Calcium-dependent signaling through CPNE1 may contribute to the pathogenesis of pSS. Moreover, the presence of significantly upregulated proteins in pSS patients’ tear fluid associated with T cell receptor signaling and Fc receptor signaling indicates alterations in calcium-dependent immune responses [33].

#### 4.4.7. Cell Adhesion

In the tear fluid of pSS patients, tear protein D52 (TPD52) was identified as an upregulated biomarker involved in cell adhesion. Additionally, LIM domain only protein 7 (LMO7) was found to be upregulated, which also plays a role in cell adhesion. Dysregulation of these proteins may contribute to the disruption of cell adhesion processes, potentially leading to the ocular manifestations observed in pSS [32,33,34].

#### 4.4.8. Cell Signaling

Several upregulated proteins in tear fluid from pSS patients are involved in cell signaling. LIM domain only protein 7 (LMO7) and E3 ubiquitin-protein ligase HUWE1 (HUWE1) are examples of proteins playing crucial roles in this process. Understanding the changes in these signaling pathways may offer insights into the pathogenesis of pSS and its differentiation from other diseases [33,34].

#### 4.4.9. Oxidative Stress

The literature review highlighted proteins associated with oxidative stress in pSS patients. For instance, APEX1 was significantly upregulated in tear fluid from pSS patients and is an enzyme activated in response to oxidative stress. This finding suggests a potential role of oxidative stress mechanisms in the pathogenesis of pSS, which could contribute to changes in tear composition and disease manifestations [32,34]. S100, a protein that reports damage due to oxidative stress, was found to be a biomarker for SS [28]. This marker has high potential to reveal the mechanism of SS onset, though further research is needed to achieve this [28].

#### 4.4.10. Metabolites (Phospholipids, Amines, Amino Acids)

Metabolites in tears have been investigated as potential biomarkers for SS and may offer insights into its underlying pathophysiological processes. One study demonstrated that the metabolic signature of tears composed of nine metabolites could differentiate newly diagnosed pSS patients from patients suffering from other causes of dryness syndrome [35].

The metabolic signature included two amino acids (serine and aspartate), one biogenic amine (dopamine), and six phospholipids (LysoPC C18:1, C18:2, C16:1; SM C16:0, C22:3; and PC aa C42:4) [35]. The dysregulation of these metabolites may reflect alterations in lipid metabolism, inflammation, and neurotransmitter signaling on the ocular surface in SS [35].

The altered phospholipids, such as LysoPC, have been associated with inflammation and pro-inflammatory cytokines (TNF-α and IL-1β), known to be implicated in SS [35]. Changes in amino acids and biogenic amines may indicate the involvement of neurotransmitter signaling and neurotransmitter-related pathways in SS [35]. The identification of metabolites in tears may serve as potential biomarkers for SS diagnosis and provide valuable information about disease pathogenesis.

### 4.5. Inflammatory Markers

#### 4.5.1. EGF

Epidermal growth factor (EGF) is a growth factor involved in epithelial wound healing and maintaining ocular surface homeostasis [31]. EGF has been detected in tears, and its levels have been found to be altered in SS patients [31]. One study reported decreased levels in DED patients with SS in comparison to those without SS [31]. On the other hand, DED patients with corneal epithelial fibrosis, meibomian gland dysfunction (MGD), and meibomian gland orifice metaplasia had increased levels [31]. This downregulation of EGF may indicate the presence of an inflammatory response and attempts at tissue repair in SS.

The roles of EGF in promoting cell proliferation and wound healing suggest its potential involvement in the restoration of damaged ocular surface tissues in SS. The dysregulation of EGF in tears may reflect the ongoing tissue remodeling processes occurring on the ocular surface in response to inflammation and immune-mediated damage.

#### 4.5.2. Mucins

Tear biomarker MUC5AC, a gel-forming mucin secreted by conjunctival goblet cells, has been investigated as a potential diagnostic tool for SS-related dry eye. A single previous study demonstrated a reduction in MUC5AC levels in tears of SS patients, but without sufficient clinical information or correlations with ocular surface parameters [29]. The association between reduced MUC5AC and the severity of dry eye or underlying SS remains unclear, due to the absence of a non-SS dry eye control group [29]. Nevertheless, a subsequent study confirmed the correlation between conjunctival lissamine green staining and SS dry eye diagnosis and found that a significant reduction in tear MUC5AC was accompanied by increased tear IL-8 levels in SS patients [29]. These findings suggest that MUC5AC, along with IL-8, could be potential surrogate markers for inflammation of the ocular surface related to SS [29].

In both SS and non-SS dry eye groups, a reduction in tear MUC5AC was observed, but the tear cytokine profile differed between the groups [29]. The non-SS dry eye group showed significant changes in IL-1b, IL-10, and TNF-a, but not IL-8. Conversely, SS patients exhibited distinctly increased tear IL-8 levels, which were significantly different from those of non-SS dry eye patients and correlated with the diagnosis of SS [29]. Increased tear IL-8 is associated with chronic autoimmune pathologies, and its secretion is induced by inflammatory cytokines, including IL-17, known to be linked to SS [29]. Hence, the elevated IL-8 levels in SS patients might reflect the underlying autoimmune pathology [29]. The results suggest that tear MUC5AC, in conjunction with IL-8, could serve as a promising candidate for SS diagnosis and warrant further investigation in larger patient cohorts [29].

Mechanistically, in murine studies, interferon-gamma (IFN-g) and tumor necrosis factor-alpha (TNF-a) have been reported to inhibit MUC5AC secretion in goblet cells stimulated with a cholinergic agonist, leading to decreased tear MUC5AC levels [29]. The reduction in goblet cell number and area in SS and non-SS dry eye patients might be attributed to the presence of inflammatory mediators in the conjunctiva and tears, or the lower tear volume in SS compared to non-SS dry eye patients [29]. Additionally, the study suggests that goblet cell area may be a better measure of goblet cell disease and dysfunction than goblet cell number in dry eye conditions, although further validation is required [29]. Collectively, these findings emphasize the significance of MUC5AC in maintaining tear film integrity and its potential as a relevant biomarker for conjunctival goblet cell function and SS-related dry eye [29,36]. 

#### 4.5.3. RNA

A recent study investigating tear biomarkers in patients with SS revealed altered expression levels of several miRNAs compared to healthy controls. Among the miRNAs, four were significantly upregulated (miR-16-5p, miR-34a-5p, miR-142-3p, and miR-223-3p), while ten were significantly downregulated (miR-30b-5p, miR-30c-5p, miR-30d-5p, miR-92a-3p, miR-134-5p, miR-137, miR-302d-5p, miR-365b-3p, miR-374c-5p, and miR-487b-3p) in SS patients. Many of these miRNAs have known roles in regulating inflammatory pathways, with the miR-16, miR-223, and miR-142 families being associated with immune cell development and differentiation [26]. Elevated levels of miR-223-3p were found to be linked to increased pro-inflammatory cytokines in the corneas of mouse models, while miR-142-3p was associated with altered intracellular signaling in salivary gland cells [26]. These findings suggest that these dysregulated miRNAs may play a critical role in the pathogenesis of SS, and targeting them could be a potential avenue for therapeutic intervention [26]. However, further investigations are necessary to fully elucidate their regulatory involvement and clinical significance [26].

miR-34a, another dysregulated miRNA found in the tears of SS patients, has been associated with regulatory networks in T cell activation [26]. Its overexpression in T cell receptors has been linked to decreased killing capacity and suppression of the NF-κB signaling process through a feedback loop. Considering the increased activity of CD4+ and CD8+ T cells in SS patients, dysregulation of the miR-34a pathway may contribute to the damage observed in glandular epithelial cells due to severe cytotoxic CD8+ T cell infiltration [26]. Additionally, miR-92 downregulation was found to be associated with the overexpression of innate defense genes in human corneal epithelial cells in response to excessive inflammation [26]. In minor salivary gland tissue samples from SS patients, miR-92a expression showed an inverse correlation with disease severity [26]. However, murine models with elevated miR-17-92 cluster expression developed autoimmune-like manifestations with enhanced proliferation of both T and B cells [26]. This highlights the complexity of miRNA regulation and its potential dual role in modulating the immune response in SS [26].

The miR-30 family, which acts as a negative regulator of B-cell activating factor (BAFF), was found to be downregulated in the tears of SS patients [26]. Since BAFF is involved in B-cell maturation, proliferation, and survival, the downregulation of miR-30 may contribute to the observed autoimmune-like manifestations in SS [26]. Excessive BAFF expression has been previously observed in B lymphocytes infiltrating the salivary glands of primary SS patients [26]. Furthermore, high levels of mRNA for SP-1 were detected in the lacrimal and submandibular glands of mice, although the human homologue of SP-1 remains unknown [24]. While these findings shed light on the potential involvement of miRNA and mRNA dysregulation in the pathogenesis of SS, further research is necessary to unravel the underlying molecular mechanisms and validate these biomarkers’ diagnostic and therapeutic potential. Nonetheless, the study of miRNAs and mRNAs as tear biomarkers holds promise due to their stability, resistance to degradation, and cost effectiveness, making them attractive candidates for clinical use in the diagnosis and differentiation of SS from other similar diseases [24,26]. 

#### 4.5.4. MMP, Interleukins, and Additional Markers

Matrix metalloproteinases (MMPs) are considered meaningful biomarkers in tear fluid. A study comparing the tear fluid from pSS patients with that of non-SS dry eye subjects revealed upregulated protein metabolism involving MMP8, SERPINB5, RPLP2, CSTB, and CST3 in pSS patients compared to controls. Additionally, erythrocyte band 7 integral membrane protein (STOM), Annexin A4 (ANXA4), and Annexin A11 (ANXA11) were significantly higher in pSS patients when compared to healthy controls. Proteomic patterns in SS patients with dry eye syndrome have identified proteins involved in host defense, immune response, inflammation, and apoptosis, including a-2-HS-glycoprotein, coagulation factor II, transferrin, orosomucoid, apolipoprotein A-II, elastase 2, serpin peptidase inhibitor, clusterin, keratin 1, C3, and 4A. Furthermore, interleukin levels such as IL-1a, IL-1b, IL-6, IL-8, TNF-a, and MMP-9 have been detected in Sjogren’s syndrome patients. These findings support the involvement of various inflammatory markers in the pathogenesis of SS and dry eye diseases [16,27,31]. Similarly, the presence of antigen-presenting dendritic cells in the central cornea has been demonstrated as a potential biomarker for systemic immune diseases in individuals with dry eye symptoms [37].

Canonical pathway analysis of tear proteins in SS patients revealed pathways correlated with stress-related signaling and inflammation in both pSS and sSS groups. Protein synthesis was identified as the top associated functional pathway. Additionally, the proteomic profiling of pSS patients showed upregulated cellular processes related to retina homeostasis and other central innate and adaptive immune responses. The pathogenesis of SS involves an autoimmune process targeting lacrimal and salivary glands, due to the presence of circulating antibodies or inflammatory mediators released from the lacrimal glands into the tears [27,31]. The alterations in tear composition and biomarkers provide valuable insights into the underlying mechanisms of SS and other similar diseases, contributing to the diagnosis and differentiation of these conditions.

## 5. Conclusions

This literature review provides a comprehensive understanding of the diagnostic potential and underlying mechanisms behind tear biomarkers in SS. The biomarkers span antimicrobial, immune regulation, and metabolic processes, as well as inflammatory markers, including matrix metalloproteinases and interleukins. Noteworthy findings include the potential diagnostic values of tear lysozyme, lactoferrin, dinucleoside polyphosphates, cathepsin, defensin, antibodies, epidermal fatty acid-binding protein, HLA-DR, ADAM10, aquaporin 5, and various miRNAs and mRNAs. These biomarkers reflect alterations in ocular surface homeostasis, immune dysregulation, and inflammatory responses in SS. These markers emphasize the diversity and complexity of SS pathogenesis and may act as targets for future research and clinical applications. This research provides insights that could guide clinicians in improving the accuracy and efficiency of SS diagnosis, improving patient outcomes by providing more timely diagnosis and intervention. Further research is required to confirm the diagnostic efficacy of these biomarkers. Overall, this review provides a valuable synthesis of existing knowledge, paving the way for future advancements in the diagnosis and management of SS.

## Figures and Tables

**Figure 1 ijms-25-10380-f001:**
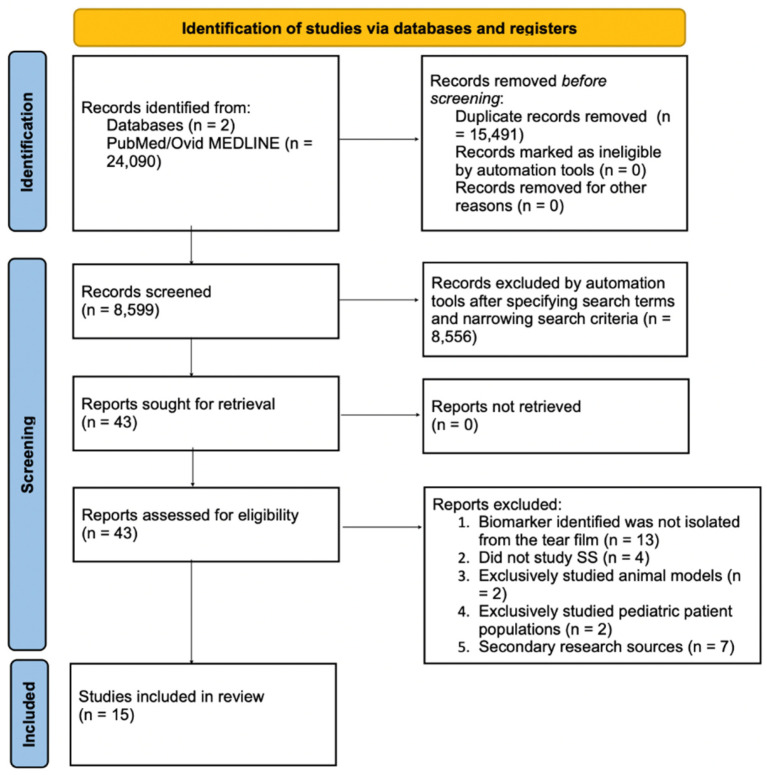
PRISMA flow diagram of the search.

**Table 1 ijms-25-10380-t001:** Key characteristics of the 15 studies included.

Author	Year	Country	Study Type	Population *
Hsiao et al.	2022	Taiwan	Cross-sectional, Case–control Observational Study	36
Akpek et al.	2020	USA	Case–control Observational Study	62
Berra et al.	2021	Argentina	Cross-sectional Observational Study	90
Levine et al.	2021	USA	Retrospective Study	128
Hamm-Alvarez et al.	2014	USA	Cross-sectional Observational Study	278
Aqrawi et al.	2018	Norway	Cross-sectional Observational Study	27
Urbanski et al.	2021	France	Cohort Study	90
Aqrawi et al.	2019	Norway	Cross-sectional Observational Study	35
Khimani et al.	2020	USA	Comparative Study	28
Brignole-Baudouin et al.	2017	France	Cross-sectional Observational Study	311
Aqrawi et al.	2017	Norway	Experimental Study	22
Karns et al.	2011	USA	Comparative Study	4
Shinzawa et al.	2018	Japan	Observational Case Series	17
Hargreaves et al.	2019	Switzerland	Case–control Observational Study	28
Edman et al.	2018	USA	Cross-sectional Observational Study	156

* Patient population is based on the number of patients tears were collected from. It may be different from the total patient population of the study; i.e., some patients only had saliva or serum collected from them.

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
