# Peer review of "A Review of the Tear Film Biomarkers Used to Diagnose Sjogren’s Syndrome"

_ijms, 2024, doi:10.3390/ijms251910380_

Round 1
Reviewer 1 Report
Comments and Suggestions for Authors
This review provides an overview of potential tear film biomarkers to diagnose Sjörgen’s syndrome.
Major concerns:
In my opinion, there are some major missing points:
1) The link between Sjörgen’s syndrome and Dry Eye Disease is insufficiently described. Moreover, there are missing references, e.g. https://doi.org/10.3390/ijms241814091
2) Information about the methods used for the identification of tear biomarkers in the different studies is missing.
Introduction: Lines 59-71: There is a general description about the tear film that has to be placed before mentioning the objectives (line 52).
Minor concerns:
Line 38: «mainly» in stead of «Mainly»
Line: 56: review about Dry Eye Disease, references are missing
Line: 65: “often” instead of “Often”
Line 141: “body fluids” instead of “bodily fluids”
Comments on the Quality of English LanguageThere are some miswording that have to be reformulated:
· Line 191: to determine its utility fully
· Line 250: reformulate the whole sentence
· Line 273: It was found that
· Line 296: reformulate the whole sentence
· Line 312: provide insights into (instead of “insights for”)
· Line 348: Although more research is required, …
· Line 494: this review contributes to.. or this review provides…
Author Response
Comment 1: The link between Sjörgen’s syndrome and Dry Eye Disease is insufficiently described. Moreover, there are missing references, e.g. https://doi.org/10.3390/ijms241814091
Response 1: First of all, thank you for reviewing my manuscript! I added a portion to the Introduction detailing the connection between Sjörgen’s syndrome and Dry Eye Disease (See "This is especially relevant... potential diagnostic and therapeutic targets."). This included the source you included (Schlegel et al., 2023).
Comment 2: Information about the methods used for the identification of tear biomarkers in the different studies is missing.
Response 2: A section was added to the Materials and Methods section further elaborating on the process I used to identify the specific tear biomarkers that were discussed in the Results and Discussion section ("The 15 papers selected for the final analysis... collection techniques and assay sensitivities.").
Comment 3: Introduction: Lines 59-71: There is a general description about the tear film that has to be placed before mentioning the objectives (line 52).
Response 3: A brief introduction to the concept of the tear biofilm was added before talking about it in more details in the subsequent paragraph ("In general, the tear film is a complex structure... microbial infections and helps repair ocular tissues.13").
Comment 4: Minor concerns and reformulation of wording
Response 4: All recommended edits were applied to my manuscript.
Reviewer 2 Report
Comments and Suggestions for Authors
This review covers the tear film biomarkers for Sjogren's Syndrome. Sections 2-5 provide detailed information for different biomarkers. However, as a systematic review, my main concern is the papers selected. Of the 22 papers used in this review, 5 were review papers, 1 was Meta-Analysis, and 1 was Systematic Review.
A systematic review is an analysis of all primary literature that exists on a specific topic. Primary literature includes only original research articles. However, narrative reviews, systematic reviews, or meta-analyses are based on original research articles, and hence are considered secondary sources. Therefore, secondary sources are not recommended for the data extraction process for your systematic review.
The alternative way is to use the original research articles cited by these sources.
Another major revision needed is on the population. For example, Pur et al. 2023 Canada paper has a population of 1058. However from this paper: "There were 1058 individuals included, with 350 individuals with dry eye, 61 with keratoconus, 43 with pterygium, 179 with meibomian gland dysfunction (MGD), 59 with graft-versus-host-disease (GVHD), 51 with Sjogren, 2 with climatic droplet keratopathy (CDK), 19 with bullous keratopathy, 2 with Fuchs’ endothelial dystrophy, 18 with vernal keratoconjunctivitis (VKC), 12 with various indications for penetrating keratoplasty, and 237 healthy controls, as well as 5 myopic and 20 diabetic individuals as comparator groups." Only 51 with Sjogren.
So suggest to double check the population.
Author Response
Comment 1: A systematic review is an analysis of all primary literature that exists on a specific topic. Primary literature includes only original research articles. However, narrative reviews, systematic reviews, or meta-analyses are based on original research articles, and hence are considered secondary sources. Therefore, secondary sources are not recommended for the data extraction process for your systematic review.
Response 1: Thank you for your review! I ended up removing the reviews, systematic review, and meta-analysis (5 papers, 1 paper, and 1 paper, respectively) from the analysis in the Results and Discussion section. However, I still cited them at times to support points made by the other 15 'Primary source' papers included in this review.
Comment 2: Another major revision needed is on the population. For example, Pur et al. 2023 Canada paper has a population of 1058. However from this paper: "There were 1058 individuals included, with 350 individuals with dry eye, 61 with keratoconus, 43 with pterygium, 179 with meibomian gland dysfunction (MGD), 59 with graft-versus-host-disease (GVHD), 51 with Sjogren, 2 with climatic droplet keratopathy (CDK), 19 with bullous keratopathy, 2 with Fuchs’ endothelial dystrophy, 18 with vernal keratoconjunctivitis (VKC), 12 with various indications for penetrating keratoplasty, and 237 healthy controls, as well as 5 myopic and 20 diabetic individuals as comparator groups." Only 51 with Sjogren.
Response 2: I went back through each of the 15 included papers and fixed the populations noted in my manuscript. I also ended up also distinguishing how many healthy controls, DED patients without any comorbid conditions (including SS), pSS-DED, and sSS-DED patients there were as an aggregated whole from the 15 included papers.
Round 2
Reviewer 1 Report
Comments and Suggestions for Authors
I have no further objections.
Reviewer 2 Report
Comments and Suggestions for Authors
The authors did a great job of addressing the suggestions.